# Study of Prepared $\alpha$-Chymotrypsin as Enzyme Nanoparticles and of Biocatalytic Membrane Reactor

**Imre Hegedüs [1,2], Marta Vitai [1], Miklós Jakab [3] and Endre Nagy [1,*]**

[1] Faculty of Engineering, Research Institute of Biomolecular and Chemical Engineering, University of Pannonia, H-8201 Veszprém, P.O. Box 158, Hungary; hegedus@mukki.richem.hu (I.H.); vitai@mukki.richem.hu (M.V.)

[2] Department of Biophysics and Radiation Biology, Semmelweis University, Tüzoltó u. 37–47, H-1094 Budapest, Hungary

[3] Engineering Research and Development Centre, Department of Material Science and Engineering, Faculty of Engineering, University of Pannonia, H-8201 Veszprém, P.O. Box 158, Hungary; jakab.miklos@mk.uni-pannon.hu

* Correspondence: nagy@mukki.richem.hu

**Abstract:** Biocatalytic kinetic effect of $\alpha$-chymotrypsin enzyme has been investigated in its free and pretreated forms (it was covered by a very thin, porous polymer layer, called enzyme nanoparticle) as well as its immobilized form into pores of polysulfone/polyamide asymmetric, hydrophilic membrane. Trimethoxysilyl and acrylamide-bisacrylamide polymers have been used for synthesis of enzyme nanoparticles. Applying Michaelis-Menten kinetics, the $K_M$ and $v_{max}$ values of enzyme-polyacrylamide nanoparticles are about the same, as that of free enzyme. On the other hand, enzyme nanoparticles retain their activity 20–80 fold longer time period than that of the free enzyme, but their initial activity values are reduced to 13–55% of those of free enzymes, at 37 °C. Enzyme immobilized into asymmetric porous membrane layer remained active about 2.3-fold longer time period than that of native enzyme (at pH = 7.4 and at 23 °C), while its reaction rate was about 8-fold higher than that of free enzyme, measured in mixed tank reactor. The conversion degree of substrate was gradually decreased in presence of increasing convective flux of the inlet fluid phase. Biocatalytic membrane reactor has transformed 2.5 times more amount of substrate than the same amount of enzyme nanoparticles and 19 times more amount of substrate than free enzyme, measured in mixed tank reactor.

**Keywords:** $\alpha$-chymotrypsin; enzyme nanoparticles; acrylamide-bisacrylamide random copolymer; organic/inorganic hybrid polymer; biocatalytic membrane reactor; polysulfone/polyamide membrane; bioreaction kinetics; N-acetyl-L-tyrosine ethyl ester; enzyme stability

## 1. Introduction

Industrial application of enzymes as biocatalysts has been highly increased during the last few decades. The estimated global market for industrial enzymes was 5.5 billion USD in 2018 and it should reach 7.0 billion USD by 2023 [1]. Enzymes usually have a relatively short lifetime (they frequently lose their activity after a few hours) and additionally they are sensitive to little changes in their micro-environment (pH, temperature, ion strange, etc.) as well as they can work effectively under optimal conditions only, therefore elongation of their lifetime is a key factor for their sustainable industrial applications.

Numerous techniques exist, which can increase enzyme catalytic stability (elongation of its lifetime and/or increase their biocatalytic activity) [2]. One of the most promising modes to reach is the enzyme immobilization, its entrapping into the internal structure of a polymer/inorganic matrix

with weak or strong bounds [3,4]. These bounds can be realized by adsorption, ionic forces or covalent linkage [3,4]. The surface/volume ratio of an enzyme carrier can be increased by reduction of the carrier's size, since small (e.g., nano-sized) carriers could be more effective than the larger ones [3,4].

Enzyme nanoparticles represent a special case of enzyme immobilization technique because every single enzyme molecule is covered with a thin, polymer layer, which is chemically bound to the surface of enzyme by some covalent linkages (multiple covalent attachments). The polymer structure of this layer, however, is flexible and does not reduce seriously the intramolecular movements of enzyme during biocatalytic processes [5–9] (Figure 1; for detailed description of synthesis, see Section 4.2). Moreover, this layer is very thin (a few nm thick) and very porous, and therefore, it allows the diffusion of substrate molecules across this layer to the active center of enzyme and also the diffusion of the product component away from the active center of the enzyme, into the bulk fluid phase [5] (see Section 4.2). However, the question arises how the biocatalytic membrane layer can affect the kinetics of the enzyme. This has not been answered until now.

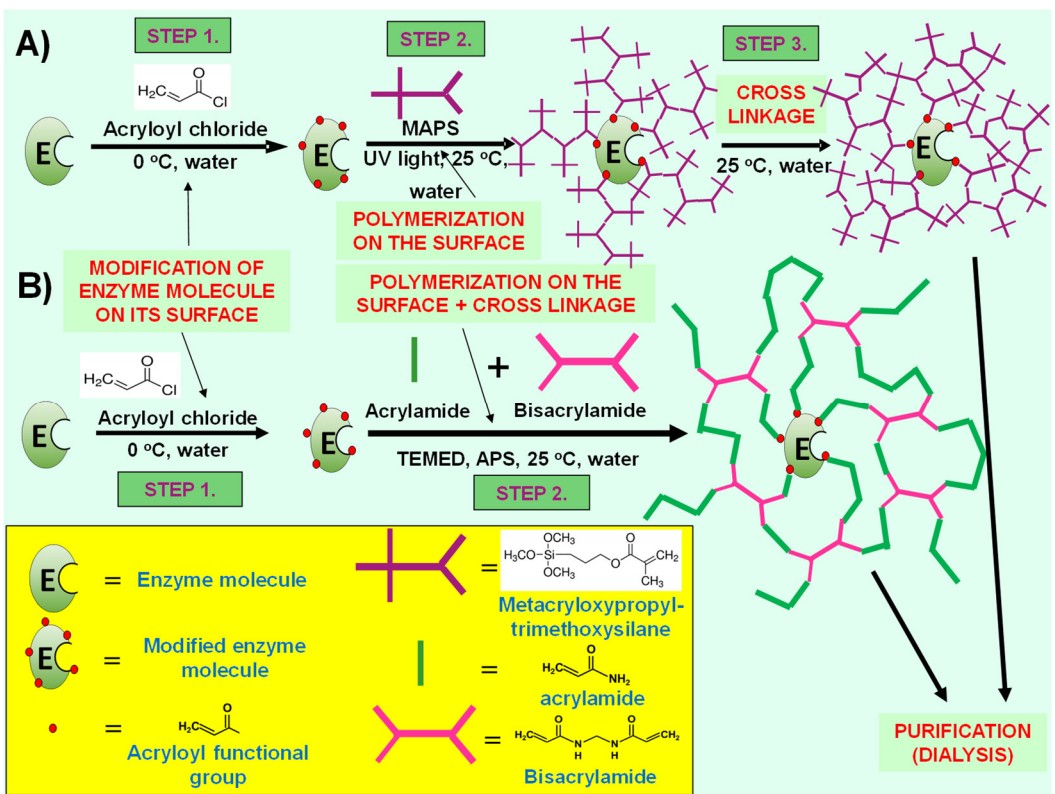

**Figure 1.** Synthesis methods and chemical structures of enzyme nanoparticles. Enzyme molecules are covered by a thin, porous polymer layer that allows the diffusion of substrate and product molecules and the intramolecular movements of molecules during the biocatalytic reaction. (**A**) Synthesis consisting of three main steps: surface modification (in water), polymerization from the surface of the enzyme (in n-hexane) and finally cross linkage between polymer fibers (in aqueous solvent). (**B**) Two-step (one-pot) synthesis of enzyme nanoparticles: modification and polymerization/cross linkage in aqueous solvent.

One of the most frequently used and intensively investigated research fields of enzymatic industrial process is the biocatalytic membrane reactors [10]. According to the estimated global market, application of the biocatalytic membrane reactors should grow from 1.9 billion USD (2018) up to 3.8 billion USD, by 2023 [11]. The biocatalytic membrane technology is widely spread in numerous industrial areas, e.g., on food and pharmaceutical industries or biofuel production [10,12].

The great advantage of the biocatalytic membrane reactors is that the biocatalytic reaction and the separation of product from the reaction mixture are simultaneous processes; therefore, the

reaction rate could be increased due to the continuous product removal, which shifts the reaction equilibrium, and with that, it improves the reaction efficiency [12–16].

Attachment of biocatalyst into or onto membrane layer could be distinguished by physical forces (adsorption of enzyme molecules to the membrane, e.g., by van der Waals interaction), by strong electrostatic forces (ionic interaction) or by covalent bound of enzymes to components of the membrane matrix (Figure 2). Membranes, applied as biocatalytic membrane reactors, could be both flat-sheet and capillary membranes. The biokinetics of the biocatalytic membrane processes are not well known yet. The conventional hydrodynamic and kinetic models can be applied for these cases as well, by adopting them to the special properties of the biocatalytic membrane reactor [12,13].

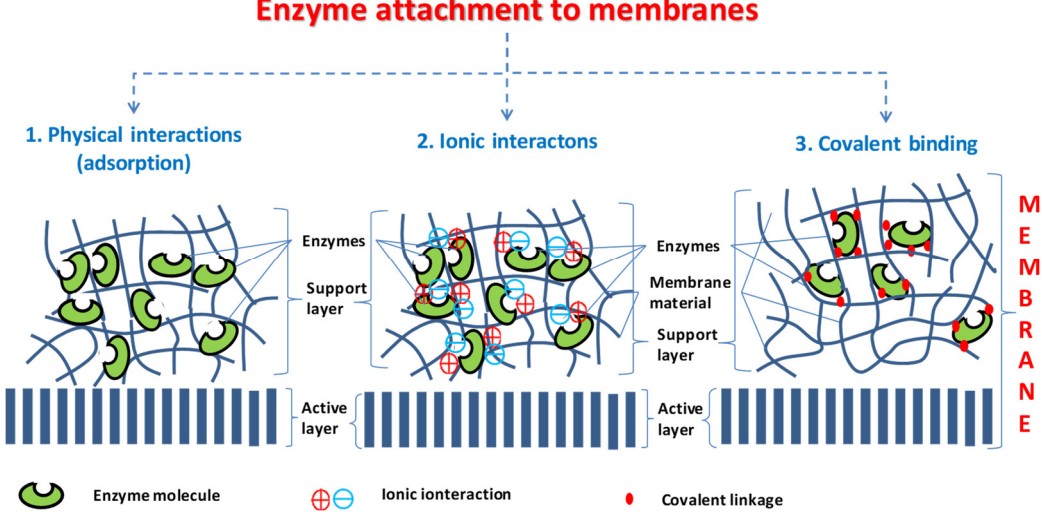

**Figure 2.** Three main types of enzyme attachment to membrane: (**1**) enzyme immobilization by physical interaction (adsorption), (**2**) enzyme immobilization by ionic linkage (electrostatic interaction), (**3**) enzyme immobilization by covalent linkage between enzyme molecules and molecules of the support material (e.g., polymer chains) of the membrane.

Authors of this study have chosen asymmetric, hydrophilic commercial flat-sheet membrane for their experiments, which is often used for wastewater treatment in industry, namely polysulfone/polyamide membrane (with molecular weight cut-off of 100 kDa), and the $\alpha$-chymotrypsin enzyme was immobilized into its porous support layer, by ultrafiltration mode (see Section 4.5). This membrane has rather thick support layer (about 40–80 μm thick) that can trap enough amount of enzyme molecules.

The $\alpha$-chymotrypsin is a well-known enzyme, and its reaction kinetics could be described well by Michaelis–Menten kinetics [17,18]. The $\alpha$-chymotrypsin enzyme is very stable; it contains two intramolecular disulfide bridges in its polypeptide chain, which can stabilize its ternary structure [17]. This enzyme keeps its biocatalytic activity for longer time than that of the commercially available ones, more or less at constant value (for several weeks at room temperature) [19]. The $\alpha$-chymotrypsin enzyme, which was used for this investigation, is a serine-protease, which is usually applied in food, pharmaceutical and wastewater industries [20,21]. Proteases successfully break down protein wastes (proteins, polypeptides, etc.) This biocatalytic process works usually at room temperature and in aqueous solutions. *N*-acetyl-L-tyrosine ethyl ester (ATEE) substrate has been chosen to experimental and kinetic investigations, because it is well soluble in water, and it does not need methanol or ethanol as additional solvent, and therefore, this enzyme can be studied in its native environment [22].

Main aim of this research is to compare the efficiency and biocatalytic mechanism of two enzyme immobilization techniques, when the enzyme is used as its free form (abbreviated: E) or as enzyme nanoparticles (as NP1: prepared by three-step synthesis method, and NP2: prepared by two-step synthesis method) and as the free (not immobilized, not pretreated as NP1 or NP2) enzyme (E) is trapped into the membrane support layer (called MI). The important question is how the

activity and stability of free enzyme and enzyme nanoparticles change in mixed tank reactor and in their immobilized forms in biocatalytic membrane reactor. The kinetics has been investigated by mixed tank reactor and biocatalytic membrane reactor. Kinetic parameter values of this bioreaction are calculated from these results and compared them to each other obtained in mixed tank reactor using enzyme nanoparticles and applying free enzyme, with its immobilization into porous membrane layer, preparing biocatalytic membrane reactor.

## 2. Results

Values of the bioreaction kinetics parameters and the stability change of the differently pretreated $\alpha$-chymotrypsin enzymes were investigated, using them in well mixed tank reactor and biocatalytic membrane reactor. The enzyme was prepared as enzyme nanoparticles in two- (denoted NP2) and three-step (NP1) synthesis methods or immobilized in an asymmetric membrane layer (MI). The kinetics, stability and efficiency of the free enzyme (denoted: E) and the pretreated one were studied. These results are shown in this section.

For comparison of efficiency of the bioreaction, measured by means of enzyme nanoparticles and by biocatalytic membrane reactor, the kinetic values of bioreaction as Michaelis–Menten constant ($K_M$) and maximal velocity of biochemical reaction ($v_{max}$) have been predicted (Table 1). These values show the time period, while the different forms of enzymes retain their biocatalytic activities as well as the values of reaction kinetic parameters, under the same conditions (room temperature and neutral pH). For kinetic measurement NP2 was chosen because activity of NP1 was very low (about one order of magnitude less than activity of native enzyme; see also Table 2), and it was hard to measure its activity values at the same enzyme concentration values than those of free ones. Therefore, NP2 was selected to study its kinetic values.

*2.1. Kinetic Constants ($K_M$ and $v_{max}$) Obtained by Free $\alpha$-Chymotrypsin and as Enzyme Nanoparticle (NP2)*

Kinetic values of 1 mg/L free $\alpha$-chymotrypsin enzyme (E) and those of this enzyme, prepared as enzyme nanoparticle (NP2: synthesized by two-step reaction method), have been investigated at pH = 7.8 and at room temperature (23 °C). Substrate conversion was plotted as a function of bioreaction time (Figure 3A), and the initial reaction rates are plotted in Figure 3B, at different values of initial substrate concentration, varied between 0.1 and 2 mM.

This calculation is based on the following approximation:

$$v_O = \left( \frac{dC_S}{dt} \right)_{t=0} = \left( \frac{\Delta C_S}{\Delta t} \right)_{t=0} \tag{1}$$

where $v_0$ is the initial value of the reaction rates (at t = 0 time) and $C_s$ is the concentration of substrate.

The determination of Michaelis–Menten parameters, values of $K_M$, $v_{max}$, was carried out by Lineweaver–Burk graphical method, both for free (not prepared, not immobilized) enzyme (E) (Figure 4) and for enzyme nanoparticles (NP2) (Figure 4). According to this method, reciprocal values of the initial reaction rates ($v_0$) are plotted as a function of the reciprocal values of the substrate concentrations. These values are correlated linearly, and the intercept of these lines on the vertical axis gets the reciprocal value of maximal velocity ($1/v_{max}$), and the cross-section with the horizontal axis can get the negative reciprocal value of the $K_M$ constant ($-1/K_M$). Value of $K_M$ constant for free E ($K_M$ = 1.2 mM) is close to data that has been measured by other authors (see Table 1. in Section 3.1). $K_M$ value of pretreated $\alpha$-chymotrypsin enzyme as NP2 ($K_M$ = 1.4 mM) is a little bit higher than that of $K_M$ value of the free enzyme. The $v_{max}$ value of NP2 is the same than that of the free E ($v_{max}$ = 0.006 mM/s).

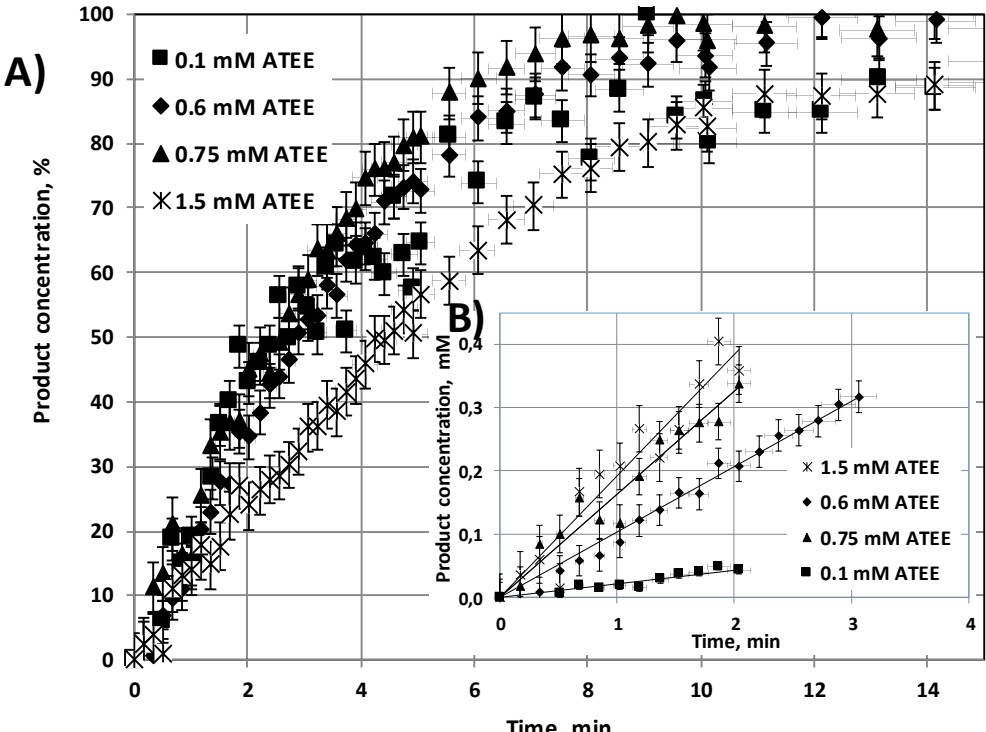

**Figure 3.** (**A**) Substrate conversion (%) plotted as a function of time; (**B**) calculation of initial reaction rate values ($v_0$) during the kinetic measurement. $v_0$ values can get as a slopes of lines which are plotted as linear regression of product concentration values as a function of time during the initial time period of biochemical reaction (e.g., during the first 2 min). Product concentration values are calculated from absorbance values of 3 mL reaction mixture in a quartz cuvette (with 1 mg/L $\alpha$-chymotrypsin and at different substrate concentrations). (Standard error values are plotted. They have varied between 4% and 6%.)

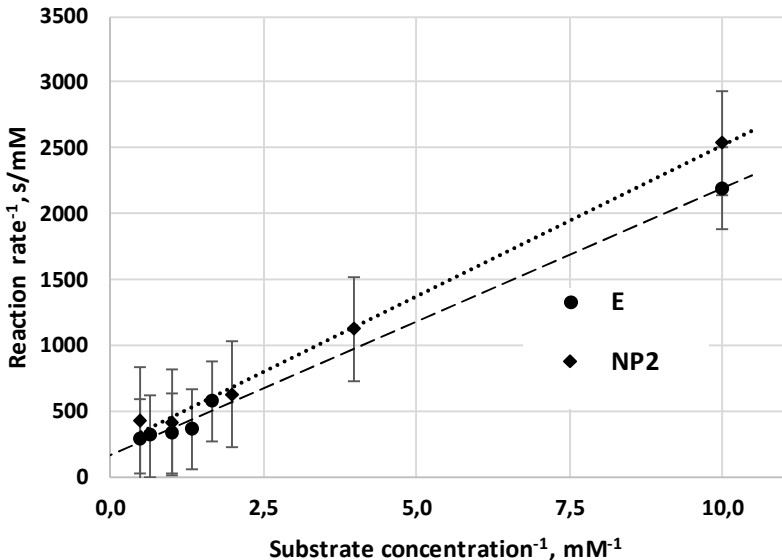

**Figure 4.** Reciprocal values of initial reaction rate ($v_0$) as a function of reciprocal substrate concentration obtained by free (not immobilized) $\alpha$-chymotrypsin enzymes (E) and enzyme nanoparticles (NP2, prepared by two-step synthesis method) plotted by Lineweaver–Burk method. ATEE was used as substrate both for E and NP2. (Standard error values of linear fitted data are plotted. They vary between ±20 and 25%. These relatively high errors are related to the graphical method applied).

### 2.2. Performance of α-Chymotrypsin Enzyme in Biocatalytic Membrane Reactor

The average pore size of polysulfone/polyamide membrane was detected by scanning electron microscopy (SEM) image, and it was found to be 160–170 nm (Figure S1; details are given in the Supplementary Materials). This pore size is essentially larger than that of the average diameter of the α-chymotrypsin enzymes (about 3–5 nm). The dense selective layer without pores hinders the adsorbed enzymes from its washing away by the flowing substrate solution, across it.

The initial reaction rate of biocatalytic reaction, in biocatalytic membrane reactor, increases by the increase of the cross convective flow rate of the substrate solution, across the membrane (Figure 5). The tendency of curves is slightly convex at every substrate concentration applied (0.5, 1 and 2 mM of ATEE substrate). When the substrate concentration is increasing, the curvatures of these curves will be slightly higher (Figure 5).

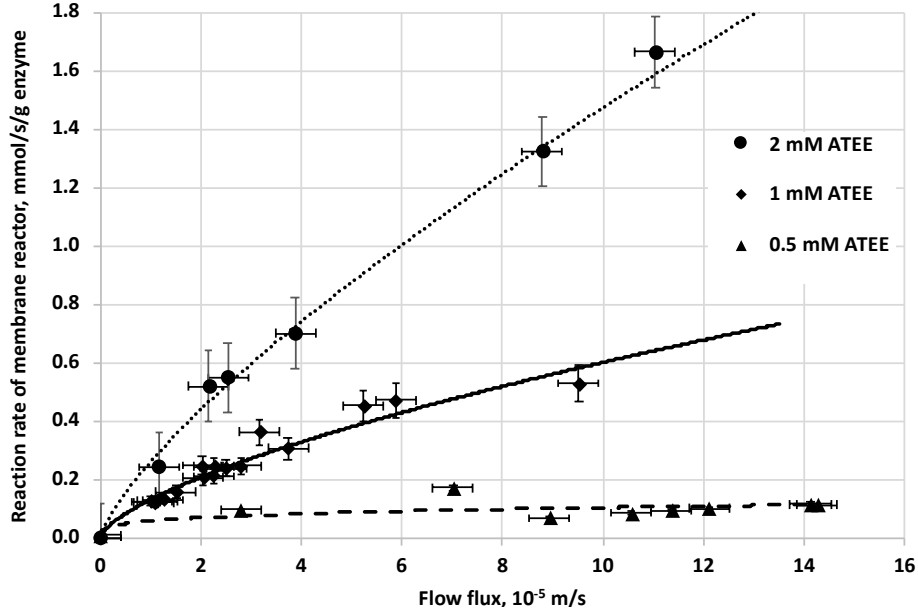

**Figure 5.** Reaction rate values of biocatalytic membrane reactor in the function of convective flow across the membrane. Results show that the reaction rate is increasing by increase of the convective flow rate across the biocatalytic membrane layer. (Standard error values are plotted. They have varied between ±4 and 6% of measured mean values.).

Contrary to the reaction rate of biocatalytic membrane reactor, the concentration of product produced by the biocatalytic reaction of the α-chymotrypsin enzyme decreases as a function of convective flow of the aqueous substrate solution ($J_w$), across the biocatalytic membrane reactor (Figure 6). The product concentration decreases dramatically between (0–3) × $10^{-5}$ m/s of $J_w$ values, and after this flow range, the slope reduces and finally the product concentration converges to a constant value (about 40%).

Curves of product concentrations, and indirectly the substrate conversion, could be described well by exponential functions with negative exponent, as a function of water flux ($J_w$), at different substrate concentrations (0.5, 1 and 2 mM of ATEE). These negative exponents describe well the concentration change as a function of the inlet fluid velocity (see predicted values (lines) and the measured data (points)). This exponential function can be derived from general mass transport equation of substrate molecule defined by means of mass transport through the biocatalytic membrane reactor (see Section 3.2).

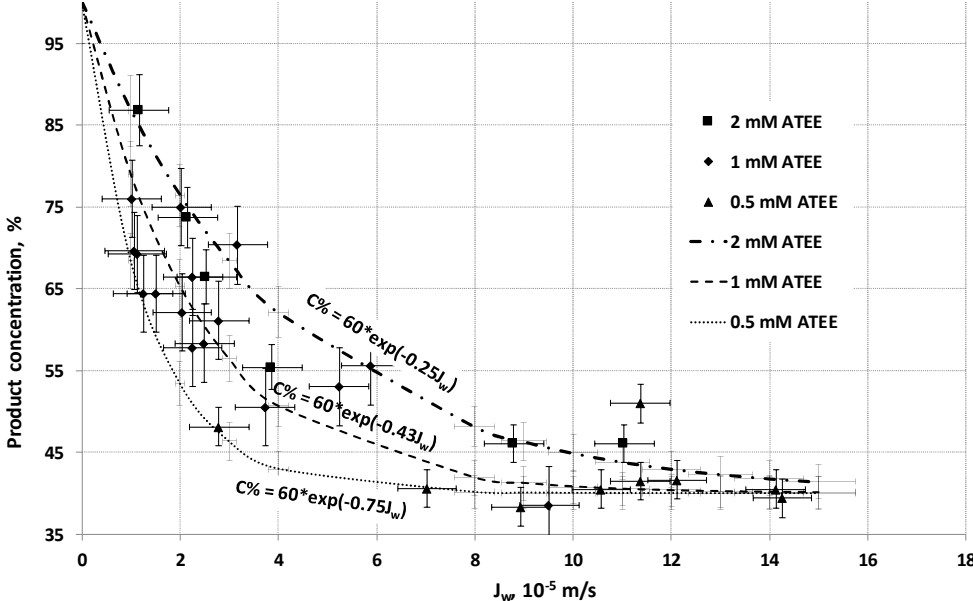

**Figure 6.** Change of the product concentration of enzyme catalytic membrane bioreactor (%) in the function of convective flow rate ($J_w$), applying different inlet substrate concentrations (0.5, 1 and 2 mM ATEE). All curves could well describe exponential functions with negative exponent (see continuous, tagged and dotted lines as calculated values) (Product concentration in % [$C_p$] can be calculated as: $C_p = C_{so} - C_s$, where $C_{so}$ is the substrate concentration at the starting time, i.e., at $t = 0$), while $C_s$ is that at $t = t$. (Standard errors values are plotted. They have varied between ±4 and 6% of measured mean values.).

### 2.3. Stability of Free and Immobilized Enzyme

One type of enzyme nanoparticles' preparation needs three main reaction steps (Section 4.2). This three-step method creates thin poly(methacryloxypropyltrimethoxysilane) hybrid, organic/inorganic polymer layer, around enzyme molecule (NP1) (Section 4.2). The other one, the two-step method, is simpler; it is synthesized poly(acrylamide-bisacryilamide) random copolymer nano-layer around the enzyme molecules (NP2) (Section 4.2).

Residual activities of native enzyme and NP1, NP2 enzyme nanoparticles were measured in mixed-tank reactor, stirred with 150 rpm, at the optimal working temperature of the enzyme (37 °C) (Figure 7). Activity change of free (not immobilized) enzymes in a function of time (enzyme inactivation) can be described by a first order kinetics [23]:

$$A(t) = A_o e^{-\lambda t} \tag{2}$$

where $A(t)$ represents the biocatalytic activity of an enzyme at time $t$, $A_0$ is the initial activity of enzyme (at $t = 0$ time), and $\lambda$ is a decay constant. Half-life time of enzyme ($t_{1/2}$) can be defined as time period over which activity of enzyme reduces to half of its original value ($A_0$):

$$A(t_{1/2}) = \frac{A_o}{2} = A_o e^{-\lambda t_{1/2}} \tag{3}$$

$$\ln\left(\frac{1}{2}\right) = -\lambda t_{1/2} \tag{4}$$

$$\lambda = -\frac{\ln\left(\frac{1}{2}\right)}{t_{1/2}} \tag{5}$$

where $\lambda$ is a decay constant.

Activity curves of all immobilization types (NP1, NP2, MI) (Figures 7 and 8), plotted as a function of time; one can suppose that these activity curves can be described well by the same

exponential formula as that used for description for the activity change of free enzyme as a function of time Equation (2). Stability of enzymes can be compared by their half-life time ($t_{1/2}$) values. Half-life time of enzyme is correlated linearly with enzyme stability. Results, in Figure 7, show that both types of enzyme nanoparticles (NP1 and NP2) retain their activity values for longer time period than that of free enzyme (E). Half-life time of free enzyme is about 0.5 h (see E on Figure 8), while that of NP2 is about 10 h long, while the NP1 has the longest half-life time, about 42 h as they are plotted in Figure 7.

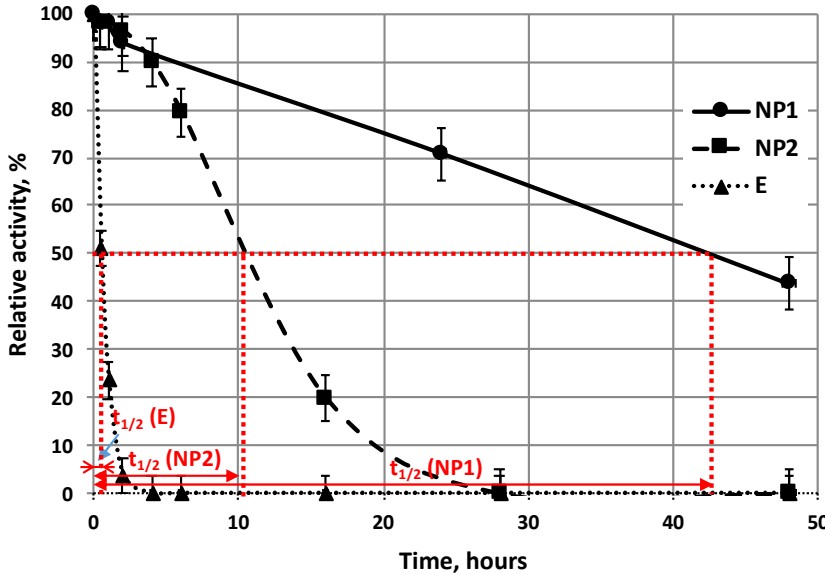

**Figure 7.** Activity changes of $\alpha$-chymotrypsin enzyme nanoparticles covered by acrylamide-bisacrylamide random copolymer (NP2) or covered by poly(methacryloxypropyltrimethoxysilane) hybrid organic/inorganic polymer (NP1) and of free (not pretreated) enzyme (E) as a function of time measured in stirred tank reactor, with 150 rpm, at 37 °C. (Standard error values are plotted. They have varied between ±3.7% up to ±5.5%.).

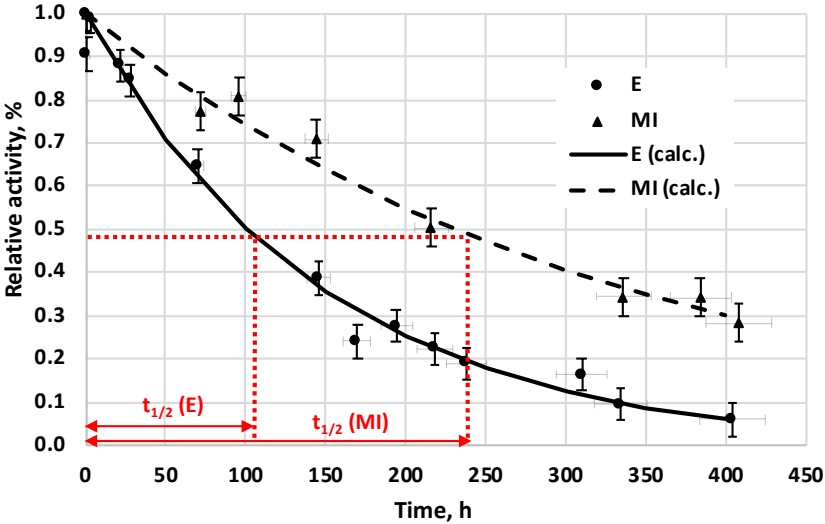

**Figure 8.** Reduction of activity of immobilized $\alpha$-chymotrypsin enzyme into porous support layer of membrane (MI) compared to activity change of the not immobilized enzyme (E) measured at the same temperature (23 °C). Half-life times of E and MI were predicted from a scale and decay constants ($\lambda$) of E and MI, using Equations (2)–(5). These calculated relative activity values are continuous line (E) and dash-line (MI). The measured (points) and the predicted values agree excellently well. (Standard error values are plotted. They have varied between ±3.8% up to ±4.4%.).

Stabilities of free enzyme (E) and its immobilized form into asymmetric, porous membrane, preparing biocatalytic membrane reactor (MI), were also compared to each other, using their activity change, at room temperature (23 °C) (Figure 8). The 0.1 mg/mL of the native $\alpha$-chymotrypsin enzyme solution was kept at 23 °C (without stirring), and its residual activity was measured as a function of time. The activity change of the immobilized enzyme (MI) was calculated by means of the outlet substrate concentration change, measured at different time points, applying the same inlet substrate concentration values and the same operating conditions (substrate cross flow rate, temperature, etc.) A flow-through biocatalytic membrane reactor was used for these measurements, in its steady-state condition. The reacted amount of substrate was then applied for prediction of the activity change of the immobilized biocatalyst. Relative activity of MI was calculated from these measured data and relative activity of MI as a function of time is plotted on Figure 8. Results show that half-life time of free E (the time period while its activity reduced to half of its initial value) lasts about 100 h while that of the MI needed about 230 h for it, which is about 2.3 times longer than that of free enzyme (Figure 8).

## 3. Discussion

Values of the reaction kinetic parameters for free enzyme and enzyme nanoparticles, measured in mixed tank reactor, are compared to those of literature data, and kinetic data of biocatalytic membrane reactor as well as the reaction efficiencies of the pretreated enzymes, comparing them to that of free enzyme, are analyzed in this section.

### 3.1. Kinetic Constants of Free Enzymes and Enzyme Nanoparticles

$K_M$ constant of immobilized $\alpha$-chymotrypsin enzyme (NP2) has more or less the same value (1.4 mM) than that of free E (1.2 mM) (Table 1). These results suggest that the thin polymer layer around NP2 does not affect seriously the affinity of $\alpha$-chymotrypsin enzyme. The slope of curve of enzyme nanoparticles obtained by Lineweaver–Burk graphical method is also about the same as that of free enzyme (Figure 3). However, the $v_{max}$ value of NP2 is also the same as that of $v_{max}$ value of free enzyme. These facts (the same $K_M$ and $v_{max}$ values) mean that the very small amount of polymer bounded to enzyme does not change the kinetics of enzyme, but the stability of enzyme can increase almost by one order of magnitude [23].

Enzyme stabilization as enzyme nanoparticle is a unique method that composes the advantages of enzyme encapsulation and covalent immobilization of enzymes in a thin, porous polymer layer [10–14]. Results obtained are compared to kinetic values found in the literature, for all, free [24] encapsulated [25] and covalently immobilized [26] $\alpha$-chymotrypsin enzymes (see $K_M$ values in Table 1, which are already published in the literature). According to the literature data, the predicted $K_M$ values were changed from 0.23 to 2.6 mM values, using the same substrate (ATEE) under more or less the same conditions (pH = 7.0–7.8 and 23–25 °C temperature) (see Table 1) [18,24,25].

**Table 1.** Maximum reaction rate values ($v_{max}$) and Michaelis–Menten constant ($K_M$) values of free (not immobilized) $\alpha$-chymotrypsin enzyme (E) and enzyme nanoparticles (NP2) comparing them with published data in the literature. The biocatalytic reaction conditions are also given (n.d. means no data). (PBS=Phosphate Buffer Saline).

| Notes | Enzyme Form | $v_{max}$, mM/s | $K_M$, mM | Conditions | | | | Literature |
|---|---|---|---|---|---|---|---|---|
| | | | | pH | T, °C | Solvent | Buffer | |
| Measured by authors | Free E | 0.006 | 1.2 | 7.4 | 23 | water | PBS | - |
| | NP2 | 0.006 | 1.4 | | | | | |
| Have been published in the literature | Free E | - | 2.17 | 7.0 | 25 | water | PBS | [18] |
| | Free E | - | 0.7 | 7.9 | 25 | water | n. d. | [24] |
| | Free E | 0.165 | 0.23 | 7.2 | 25 | water | phosphate TRIZMA | [25] |
| | Encapsulated | 0.024 | 1.64 | | | | | |
| | Free E | - | 0.73 | 7.3 | n.d. | water | Phosphate | [26] |
| | Immobilized in agarose gel | - | 1.7-2.6 | | | | | |

Shapiro and Pykhteeva (1998) encapsulated $\alpha$-chymotrypsin enzyme in liposomes, and they measured its kinetic values [25]. $K_M$ value of encapsulated enzyme was increased up to eight times (1.64 mM) compared to that obtained by the free enzyme (0.23 mM) (Table 1). Values of $v_{max}$ of immobilized enzyme obtained were about eight times less (0.024 mM/s) than that of free enzyme (0.165 mM/s) (Table 1). These results can suggest that enzyme immobilization as enzyme nanoparticles does not reduce $v_{max}$ value as much as encapsulation of enzyme in liposomes, and $K_M$ value is not increased as much as that by liposomes.

Clarck and Bailey (2002) immobilized bovine $\alpha$-chymotrypsin enzyme covalently into CNBr-Sepharose 4B gel [26] and about 2.3 times higher $K_M$ value was measured in the case of immobilized enzyme (1.7 mM) than that of free enzyme (0.73 mM). The covalent linkage was realized between primary amino groups of enzyme molecule and functional groups of gel matrix, similarly to enzyme nanoparticles prepared by authors. $K_M$ value of enzyme, immobilized into Sepharose gel, under different conditions, increased to about 1.7–2.6 mM. This value is 2.3–3.5 times higher than the $K_M$ value of free enzyme (0.73 mM) [26] (Table 1).

Surprisingly, $K_M$ value of the $\alpha$-chymotrypsin enzyme nanoparticles (NP2) is about 1.17 times as much as $K_M$ value of free enzyme (E), and its increase is lower than that in the case of the encapsulated one (about 5 times increase) [25] or that in case of covalent immobilization (2.3–3.5-times increase) [26] (Table 1). This relatively low change of $K_M$ value of enzyme nanoparticle could be thanks to the very thin, porous polymer layer, around enzyme, whose thickness is commensurable with the size of enzyme molecule (about 3–5 nm [6]). Moreover, this polymer layer has high porosity (about 2–3 polymeric/oligomeric fibers are connected to a single enzyme molecule; see detailed description in Section 4.2 and [14]).

### 3.2. Analyzing of the $\alpha$-Chymotrypsin Enzyme Immobilized in Biocatalytic Membrane Reactor (MI)

The cutoff value of membrane used for enzyme immobilization in this study is 100 kDa, and its average pore size about 150 nm (see Section 2.2 and Figure S1). This is significantly higher than the size of $\alpha$-chymotrypsin enzyme (3 × 3 × 5 nm). This fact should mean that immobilization forces between enzyme molecules and functional groups of membrane pores could be strong enough for stable work by the biocatalytic membrane reactor.

The general equation of mass transport through biocatalytic membrane layer can be written as [27]:

$$D\frac{d^2 C_S}{dx^2} - v\frac{dC_S}{dx} - \frac{v_{max}C_S}{K_M + C_S} = 0 \tag{6}$$

where $D$ is a diffusion constant of substrate molecule, $C_S$ is the concentration of substrate molecule, $v$ is the convection velocity, across the biocatalytic membrane layer, and $v_{max}$ denotes the maximal velocity of biocatalytic reaction and $K_M$ is the Michaelis–Menten constant. Diffusion constant of substrate molecule (ATEE) falls in the range of $10^{-9}$ m²/s [23]; therefore, diffusion term of mass transport equation $\left(D\left(d^2 C_s / dx^2\right)\right)$ is rather low, and in our case, it is negligible compared to the value of convective term $\left(-v\left(dC_s / dx\right)\right)$. The value of the physical mass transfer coefficient of membrane, $D/\delta = (1 \times 10^{-9}$ m²/s$)/(200 \times 10^{-6}$ m$) = 0.5 \times 10^{-5}$ m/s. The convective velocity was changed $(1-15) \times 10^{-5}$ m/s (Figure 6); thus, these values are essentially higher than that of membrane physical mass transfer coefficient, practically in the whole convective flow regime. Accordingly, the first term in Equation (1) can be neglected. According to this consideration, the mass transport Equation (6) can be reduced to the following differential equation:

$$-v\frac{dC_S}{dx} - v_{max}\frac{C_S}{K_M + C_S} = 0 \tag{7}$$

When $C_s \ll K_M$, then $K_M + C_s \approx K_M$; the Michaelis–Menten equation could be reduced to first order reaction, and the differential mass transfer equation will be as follows:

$$-v\frac{\mathrm{d}c}{\mathrm{d}x} - \frac{v_{\max}}{K_M}C_S = 0 \tag{8}$$

$$\frac{\mathrm{d}C_S}{\mathrm{d}x} + \frac{v_{\max}}{K_M v}C_S = 0 \tag{9}$$

If $B = v_{\max}\delta/(K_M v)$ and $X = x/\delta$, then

$$\frac{\mathrm{d}C_S}{\mathrm{d}X} + BC_S = 0 \tag{10}$$

The general solution of Equation (10) is as follows:

$$C_S = Te^{-Bx} \tag{11}$$

where $T$ is an integration constant. When $x = 0$, the substrate concentration is equal to its initial value, $C_s = C_{so}$, and then, $T = C_{so}$. When $X = x/\delta$, then

$$C_s = C_{so}e^{-BX} = C_{so}e^{-(v_{\max}\delta/(K_M v))X} \tag{12}$$

Curves of exponential functions with negative exponents could be approximated well with the measured points. They describe the characteristic patterns of product concentration data of biocatalytic membrane reactor, plotted as a function of convective water flux, $v$ (see dotted lines on Figure 6).

*3.3. Comparison the Efficiency of Biocatalytic Reactions by Enzyme Nanoparticles to that by Biocatalytic Membrane Reactor*

The above-mentioned three different immobilization method of the $\alpha$-chymotrypsin enzyme (NP1, NP2, MI) can be compared to each other, when one tries to calculate how much product is maximum generated by given amount of enzyme (e.g., 1 mg enzyme as free, E, and immobilized enzymes as, NP1, NP2, or MI), (Table 2).

**Table 2.** Reduction of biocatalytic activity of $\alpha$-chymotrypsin enzyme during synthesis steps of two different synthesis methods (initial enzyme activities were considered as 100% in both cases).

| Immobilization Type | Enzyme Modification | Polymerization | Final Value |
|---|---|---|---|
| NP1 | 97% | 13% | 13% |
| NP2 | 97% | 57% | 55% |

For this reason. let us compare at first, how the initial activity values of immobilized enzymes change. Table 2 summarizes the actual activity values during each synthesis step, comparing them to the activity values before the given step in the cases of both syntheses methods of NP2 or NP1. Final activity values, after the synthesis, have been compared to the initial enzymatic activity values before synthesis.

Table 2 shows that the final value of biocatalytic activity of NP1 is only 13% of the original value of free enzyme, E, (this behavior decrease is caused by enzyme modification and synthesis of the polymer layer). Contrary to it, final activity of NP2 after its synthesis steps is about 55% of that of free enzyme. The reduction of activity in this case is mainly caused by the enzyme modification and polymerization. After all, one can conclude that two-step synthesis method for production of NP2 is more effective than the three-step one produces NP1.

For estimation how much better is the single stabilization method, one can define relative efficiency of enzyme. This relative enzyme efficiency means how many more substrate molecules are converted to product by a given amount of enzyme during a given reaction time by differently stabilized enzyme molecules.

Performance of enzymes as free E, NP1, NP2 or MI could be characterized by two factors: (a) the initial reaction rate ($v_0$), which corresponds to the initial activity of enzymes and (b) to their half-life time ($t_{1/2}$). Multiplication of these factors with each other leads to a value, which should linearly be correlated to the efficiency of a given type of prepared enzyme produced here (free E, MI, etc.) When one divides these effectivity values by the efficiency value of free E, one gets the relative efficiency ratio, which could characterize the efficiency of different enzyme stabilization methods; this value of free E is assumed to be unity, e.g., relative efficiency of NP1 is the initial reaction rate of NP1 ($v_{0(NP1)}$) multiplied by its half-life time ($t_{1/2(NP1)}$), and this value is divided by the multiplication of the same values for free E ($v_{0(E)}t_{1/2(E)}$):

$$eff_{NP1} = \frac{v_{o(NP1)}t_{1/2(NP1)}}{v_{o(E)}t_{1/2(E)}} \tag{13}$$

Similarly, one can get $eff_E$, which equals to 1, values of $eff_{NP2}$ and $eff_{MI}$ (Table 2).

According to Equation (13) effectivities of stabilization methods are comparable (Table 3), The relative values of initial reaction rates are put in the final column of Table 3. Calculation of relative activity of MI is based on its value at the highest water flow ($15 \times 10^{-5}$ m/s). According to absorbance measurement of the permeate originated by means of the highest water flux obtained by biocatalytic membrane reactor, one can conclude that the same amount of immobilized enzyme can catalyze transformation of substrate by an 8.1-times higher amount than that of free E, applying mixed tank reactor for the bioreaction.

**Table 3.** Comparison of efficiency of 1 mg $\alpha$-chymotrypsin enzyme pretreated by different methods, calculated as product of half-life times and initial activities. Free E: not immobilized $\alpha$-chymotrypsin enzyme; NP1 (three-step synthesis method) enzyme nanoparticles with trimethoxysilyl polymers; NP2 (two-step synthesis method) enzyme nanoparticles with acrylamide-bisacrylamide layer. MI: enzyme immobilized into the porous membrane layer.

| Enzyme Type | Ratio of Half-Life Times | Relative Activity, Considered with Table 2 | Efficiency Ratio |
|---|---|---|---|
| Free E | 1.0 | 1.0 | 1.0 |
| NP1 | 84 | 0.13 | 11 |
| NP2 | 20 | 0.57 | 11 |
| MI | 2.3 | 8.1 | 19 |

The efficiency ratio of NP1, NP2 and MI (their efficiency values divided by $eff_E$) are also listed in Table 3. Both types of immobilized enzymes (NP1and NP2) as enzyme nanoparticles have about 7 times higher efficiency ($eff_{NP1}$ and $eff_{NP1}$) than that of the free enzyme ($eff_E$). Surprisingly, efficiency value of enzyme immobilized into pores of membrane ($eff_{MI}$) obtained was even 19 times more, depending on the water flux, than that of the free enzyme, measured in mixed tank reactor ($eff_E$). These calculation results prove that the enzyme immobilized in biocatalytic membrane layer (MI) could work more efficiently than other promising methods (NP1, NP2), using them in mixed tank reactor.

Performance of biocatalytic membrane reactor with immobilized NP2 enzyme nanoparticles (MI-NP2) as a multi-level immobilization method will be investigated in the next future.

## 4. Materials and Methods

### 4.1. Materials and Instruments

$\alpha$-chymotrypsin from bovine pancreas, phosphate buffer saline tablet for 100 mL solution, *N*-acetyl-L-tyrozin ethyl ester (ATEE), acryloyl chloride, acrylamide, bisacrylamide, tetramethylethylene dimanine (TEMED), ammonium peroxodisulphate, dialysis tubing (MWCO 12 400 kDa, avg. flat width 32 mm), 1,3-bis[tris (hydroxymethyl)methylamino]propane or Bis-Tris propane, sodium bis(2-ethylhexyl) sulphosuccinate or aerosol OT (AOT),

methacryloxypropyltrimetoxysilane (MAPS), 2,2-azobis(2,4-dimethylvaleronitrile (Sigma-Aldrich, St. Luis, MO, USA). Tris(hydroxymethyl) aminomethane (VWR Inc., West Chester, PA, USA), HCl (37%, Carlo Erba Reagents sri, Cornaredo, Italy). Disodium hydrogen phosphate, potassium dihydrogen phosphate, calcium chloride, 2-propanol, *n*-hexane, calcium chloride (Scharlab,S.L. Barcelona, Spain).

### 4.2. Synthesis of Enzyme Nanoparticles

The polymerization step in the synthesis for preparing single enzyme nanoparticles of the α-chymotrypsin enzyme (NP1) was carried out in a double-walled stirring vessel. The solution was irradiated by a UV-lamp made by Vilber Lourmat. Filtration of the surface-polymerized enzymes (NP1) was carried out with a syringe filter (pore size 0.1 μm) made by Merck Millipore (Burlington, MA, USA). UV-spectra of feed solutions and permeates of biocatalytic membrane reactor were recorded by T80+ UV/VIS spectrophotometer made by PG Instruments LTD (Lutterworth, UK). A GFL 3031 shaking incubator (Progen Scientific, London, UK) was used for the stability measurements of free E, NP1 and NP2.

Two different types of enzyme nanoparticle synthesis were prepared and measured their performances. One of these synthesis methods contains three main steps (NP1), and the other one consists of two synthesis steps only (NP2) (Figure 1). Detailed description of the three-step synthesis of enzyme nanoparticles was published earlier [5,6]. The first step, in the three-step method, is enzyme modification from primary amine groups, on its surface (in aqueous solution). The surface-modified enzyme obtained is dissolved in n-hexane by a specific method (hydrophobic ion pairing), and then, in situ polymerization is started from the modified parts of the enzyme molecule. In this process, vinyl groups on the enzyme surface (synthesized in the first step) are well exposed to the organic solvent (and reagents). Finally, enzyme-polymer nanocomposite was dissolved in water, and polymer chains were cross-linked as a polymer layer around enzyme molecule. This layer is hybrid organic-inorganic polymer, which contains trimethoxysilylmethacrylate monomers (Figure 1A).

The other synthesis method contains two steps and realized in aqueous solution (Figure 1B). Two-step synthesis of enzyme nanoparticles was realized according to our previous publication [8]. This synthesis is a modification of the original method by Yan et al. [7]. The first step of the synthesis of enzyme nanoparticles is enzyme modification, and after it, α-chymotrypsin enzymes are dissolved in PBS buffer (pH = 7.8) to in situ polymerization, applying 0.1 g/L enzyme concentration. A typical synthesis method is the following: 20 mL of this enzyme solution was cooled down to about 0 °C, and then, 20 μl of acryloyl chloride was added to the reaction mixture. After it, the reaction mixture was slightly warmed to room temperature. The second, polymerization step is started after about 1 h of modification step. Acrylamide-bisacrylamide mixture (150 μl) (molar ratio is 9 acrylamide/1 bisacrylamide) was added to the modified enzyme solution. After it, 25 μl of tetramethylethylene diamine (TEMED) and ammonium persulphate (250 μl of 10% fresh aqueous solution) were added to this solution. The reaction mixture was stirred continuously for 24 h period, at room temperature (23 °C).

When the reaction was finished, enzyme nanoparticle solution was cleaned by dialysis tubing (using 50 kDa cutoff cellulose dialysis tube) three times.

### 4.3. Measurement of Enzyme Activity

Activity of enzyme nanoparticles, synthesized by both methods (NP1 and NP2) was measured under the same conditions. NanoDrop UV-VIS spectrophotometer (Thermo Fischer Sccientific, Waltham, MA, USA) was used for spectroscopically following the absorbance changes for the measurement of biocatalytic activity of enzyme and enzyme nanoparticle. 0.1 mg/L as typical α-chymotrypsin solution was prepared in Tris/HCl buffer (80 mM Tris buffer, pH = 7.8). ATEE substrate (2 mM) was dissolved in it, e.g., a typical reaction mixture was the following: 0.3 mL of 1 mg/L enzyme solution was mixed with 1.2 mL Tris/HCl buffer (80 mM Tris buffer, pH = 7.8) and with 1.5 mL 2 mM ATEE/Tris. The reagent solution was stirred by magnetic stirrer during the

absorbance measurement, at room temperature (23 °C). The absorbance of the 3 mL volume reaction mixture, at 237 nm of wavelength, was detected using 1.0 cm long quartz cuvette in given time-intervals (typically absorbance was measured in every 30 s for 90 min time period). Every measurement was repeated three times.

### 4.4. Measurements of Bioreaction Kinetics

Kinetic parameters of the α-chymotrypsin enzyme, taken it into a quartz cuvette of 3 mL, were also measured under the same conditions (at room temperature, 23 °C and stirring by miniaturized magnetic stirrer, with about 100 rpm). The absorbance of the investigated reaction mixture was detected continuously (NanoDrop Thermo-Fischer spectrophotometer was used for the absorbance detection). Absorbance values of enzyme-substrate mixture were measured at 237 nm wavelength as a function of time, with different initial concentration values of substrate molecules (from 0.1 mM ATEE up to 4 mM ATEE substrate). Every measurement was repeated three times. Concentration values of product were calculated used the following equation

$$C_p = \frac{I_o - I(C_S)}{\varepsilon} f(C_S) \tag{14}$$

where $I_o$ is the absorbance value of reaction mixture, at 237 nm and at the starting time, i.e., $t = 0$; $I$ notation represents the absorbance values of the reaction mixture, as a function of time and $f$ means the correction factor of absorbance values. $f$ correction factor was calculated by the actual slopes of calibration curve of substrate absorbance values plotted as function of substrate concentration ($c_S$). The initial reaction rates ($v_o$) were calculated on curves of each substrate concentrations by linear regression of the starting part of their concentration curves (typically the first two minutes) that is closely linear. Figure 2B shows some examples how $v_o$ values are calculated. Black markers symbolized the concentration values of the product considered in linear regression.

### 4.5. Membrane Characterization

Microstructure and pore size distribution of membrane were tested by FEI/ThermoFisher Apreo S scanning electron microscope (SEM) (Thermo Fisher Scientific. Waltham. MA. USA Company). This SEM investigation was realized in a low vacuum with an accelerator voltage of 5 kV. Membrane material is not conductive; therefore, a thin electron transparent layer (about 15 nm thick Au/Pd layer) was prepared on the membrane surface by JEOL IB-29510VET type evaporator (JEOL USA, Inc., Peabody, MA, USA).

### 4.6. Immobilization of α-chymotrypsin Enzyme in Membrane

Before immobilization processes, membrane was wetted (it has been placed into extra pure ion exchanged water for 24 h). After it, the asymmetric membrane layer was installed into a home-made pressurized container with temperature control (Figure S2). Support layer of membrane was faced with the feed side and active layer faced with permeate site during these processes. The virgin membrane layer was washed by extra pure ion exchanged water for three times, before the immobilization process. Membrane washing was carried out, by ultrafiltration mode, under pressure, using $N_2$ gas (Figure S3).

Immobilization of the α-chymotrypsin enzyme into the porous layer of membrane was carried out by physical absorption of enzyme molecules. Home-made pressurized membrane module was used (Figures S2 and S3), and 10 mL of 0.1 mg/mL α-chymotrypsin enzyme solution was forced from the feed side through the membrane by 0.5 bar pressure. Enzyme (1 mg) was immobilized into support layer of membrane, with $1.73 \times 10^{-3}$ m$^2$ surface of membrane, with ultrafiltration method forcing the enzyme solution through the membrane, facing the porous support layer. The enzyme content of the permeated solution, passed through the dense layer, was detected by measuring the absorbance of permeate. After the immobilization process, the enzyme containing membrane was washed by distilled water, and its absorbance was also detected. Accordingly, the biocatalysts were immobilized by physical interactions into the porous support layer of polysulfone/polyamide, and

its catalytic performance was measured by separated experiments. No measurable enzyme lost was detected in the permeate solution through the dense active membrane layer, using absorbance measurement at 280 nm.

*4.7. Investigation of Performance of Biocatalytic Membrane Reactor*

Investigation of performance of biocatalytic membrane reactor was realized by forcing of different substrate solutions through the membrane reactor, by flow-through mode (or dead-end mode). ATEE substrate was dissolved in PBS buffer (10 mM, pH = 7.8). Three different substrate concentrations (0.5, 1 and 2 mM of ATEE substrate) were used by different pressures (2, 3, 4, 5, 6, 8 and 10 bar). Permeate was collected, and its absorbance was investigated at 237 and 280 nm wavelengths for measuring the concentration of the ATEE substrate. The reaction rate was calculated by concentration change of substrate during the time interval while permeate sample was gathered.

## 5. Conclusions and Future Trends

The $\alpha$-chymotrypsin enzyme has been investigated in its free and differently pretreated forms in mixed tank reactor and in biocatalytic membrane reactor. Performances of free (not immobilized, not pretreated) and pretreated enzymes (as enzyme nanoparticles) were compared to each other. Enzyme nanoparticles are nanoconjugates, where a few nanometer thick, porous polymer layer has been synthesized around enzyme molecule. Two types of enzyme nanoparticles are prepared, namely, poly(methacryloxypropyltrimethoxysilane) hybrid organic/inorganic polymer containing enzyme nanoparticle (NP1) and acrylamide-bisacrylamide random copolymer containing enzyme nanoparticle (NP2). Enzyme has also been immobilized into pores of polysulfone/polyamide membrane (MWCO: 100 kDa) (MI) and studied.

NP1, prepared by three-step synthesis method, has 84 times higher half-life time, (obtained with 150 rpm stirring speed, at 37 °C) than that of the free enzyme (E), but its initial activity value is only 13% of activity value of free enzyme. NP2, prepared by two-step synthesis method, has about 20 times higher half-life time comparing it to that of free E, at pH = 7.8 with 150 rpm and at 37 °C, but its initial activity is about 57% of that of free enzyme. Stability of the $\alpha$-chymotrypsin enzyme immobilized into the porous layer of polysulfone/polyamide membrane was increased, and its half-life time was 2.3 times higher than that of free E, under the same conditions (pH = 7.8 and 23 °C), measured in mixed tank reactor.

Kinetic studies enable the user to compare $K_M$ and $v_{max}$ values of free E and NP2 enzyme nanoparticles. Results show that $K_M$ value of NP2 is only 1.17 times higher than that of $K_M$ of free enzyme, since $v_{max}$ values are the same in both cases (NP2 and free enzyme, E).

The same amount of MI (enzyme immobilized into asymmetric membrane layer) enzyme can catalyze about 8 times more amount of substrate than that of free E, measured in mixed tank reactor.

The efficiency of different immobilized methods of the $\alpha$-chymotrypsin has been defined as their initial reaction rate ($v_0$) values multiplied by their half-life times ($t_{1/2}$). Enzyme nanoparticles (both NP1 and NP2) are 11 times more effective than that of free enzyme, E, while MI is 19 times more effective than that of free enzyme, E, applying mixed tank reactor.

Given the novelty of the results presented, it was proved that preparation of the enzyme nanoparticles can significantly extend the active reaction time period of the enzyme, providing much longer applicability. On the other hand, the immobilization of free enzyme into porous membrane layer preparing a biocatalytic membrane reactor can essentially increase the bioreaction rate, and it can work for longer time than that of the free enzyme.

Considering the future trends, it comes to the front to extend the active lifetime in order to make the application of enzymes more economic. Two main possibilities can be emphasized: the genetic modification of the enzyme as well as the elaboration of different preparation processes producing, e.g., enzyme nanoparticles and biocatalytic membrane reactor, as well as their combinations.

**Supplementary Materials:** The following are available online at www.mdpi.com/2073-4344/10/12/1454/s1, Figure S1: Scanning electron microscopic image of pores of polysulfone/polyamide membrane (support layer) used for enzyme immobilization; Figure S2: Schematic figure of the home-made membrane device, whose temperature and pressure of the feed solution was regulated; Figure S3: Photo of home-made membrane bioreactor.

**Author Contributions:** I.H., experimental investigation, writing of the study, visualization; E.N., validation, supervision; M.V., discussion of experimental results; M.J., measurements with scanning electron microscopy. All authors have read and agreed to the published version of the manuscript.

**Funding:** This research was funded by the Hungarian National Development Agency, OTKA, with grant number K116727 and grant number GINOP-2.3.2-15-2016-00017.

**Acknowledgments:** The authors acknowledge the support for scanning electron microscopic studies that were performed at the electron microscopy laboratory of the University of Pannonia. Authors express their special thanks for the kind help of Peter Pekker in scanning electron microscopic imagination and for sample preparation.

**Conflicts of Interest:** The authors declare no conflicts of interest.

## Notations

| | |
|---|---|
| $\lambda$ | decay constant |
| $\delta$ | diameter of the membrane |
| $A_0$ | initial activity of the enzyme |
| $C_{so}$ | substrate concentration at the starting time, i.e., at t = 0 |
| $C_P$ | concentration of product |
| $C_s$ | concentration of substrate |
| $D$ | diffusion constant of ATEE substrate molecule |
| $eff_E$ | relative effectivity of E component $\quad eff_E = \left(v_{o(E)}t_{1/2(E)}\right)\Big/\left(v_{o(E)}t_{1/2(E)}\right)=1$ |
| $eff_{NP1}$ | relative effectivity of NP1 component $\quad eff_{NP1} = \left(v_{o(NP1)}t_{1/2(NP1)}\right)\Big/\left(v_{o(E)}t_{1/2(E)}\right)$ |
| $eff_{NP2}$ | relative effectivity of NP2 component $\quad eff_{NP2} = \left(v_{o(NP2)}t_{1/2(NP2)}\right)\Big/\left(v_{o(E)}t_{1/2(E)}\right)$ |
| $eff_{MI}$ | relative effectivity of MI component $\quad eff_{MI} = \left(v_{o(MI)}t_{1/2(MI)}\right)\Big/\left(v_{o(E)}t_{1/2(E)}\right)$ |
| $f$ | correction factor of absorbance values |
| $I$ | actual absorbance values of the reaction mixture |
| $I_o$ | initial absorbance value of reaction mixture |
| $J_w$ | convective flow rate of the aqueous substrate solution |
| $K_M$ | Michaelis–Menten kinetic constant |
| $t_{1/2}$ | half-life time of enzyme |
| $v$ | convection velocity |
| $v_o$ | initial velocity of the biochemical reaction |
| $v_{max}$ | maximal velocity of biocatalytic reaction |

## Abbreviations

| | |
|---|---|
| ATEE | *N*-acetyl-L-tyrosine ethyl ester |
| E | Free (not immobilized) $\alpha$-chymotrypsin enzyme |
| NP1 | Enzyme nanoparticles with poly(methacryloxypropyltimethoxysilane) |
| NP2 | Enzyme nanoparticles with poly(acrylamide-bisacrylamide) |
| MAPS | Methacryloxypropyltrimetoxysilane |
| MI | Immobilized $\alpha$-chymotrypsin enzyme into pores of asymmetric membrane |
| MWCO | Molecular weight cut-off value |
| PBS | Phosphate buffer saline |
| SEM | Scanning electron microscope |
| TEMED | Tetramethylethylene dimanine |

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
