# Peer review of "Study of Prepared α-Chymotrypsin as Enzyme Nanoparticles and of Biocatalytic Membrane Reactor"

_catalysts, doi:10.3390/catal10121454_

Round 1
Reviewer 1 Report
The present manuscript reports the immobilization of a-chymotrypsin with two different methods, membrane adsorption and formation of enzyme nanoparticles in polymer layer. The immobilization greatly enhanced the stability properties of the biocatalyst, with minimal effect on the kinetic properties of the enzyme. The overall work could present some interest in terms of biocatalysts improvement, but the authors do not describe adequately the contribution of their work in the field, and the advantages it could offer. However, the most important issue of the present manuscript is that it is very hard to read, due to the very poor use of English language. Therefore, the manuscript needs editing by a professional of native English speaker. Also, there are too many figures and tables, and their number should be reduced, probably some of them could be supplied as supplementary material.
Other issues are the following:
The title is too generic, please choose a more informative title
Fig. 9 contains grammatical errors (POLIMERYZATION)
L425-428: Please indicate the parameters of SEM analysis.
Fig. 2: Please arrange the graph label in concentration order
Fig. 3: This figure is a bit confusing. The a and b plots could be merged into one, or maintain the same scaling in y axis. Also, different substrate concentrations seem to have been used for the two enzyme preparations (free and immobilized). Why is that?
Fig. 4: What do the red numbers stand for? Please indicate in the figure.
In figure 6, the authors present the product concentration in the y axis, but in the text they refer to ‘Curve of substrate conversion as a function of water flux (Jw) at different substrate concentrations (0.5 mM, 1 mM and 2 mM of ATEE) could be described well by exponential functions with negative exponent, where these negative exponents are correlated with the reciprocal values of substrate concentrations’ (L167-169), which one is correct? Is product or substrate showed in Fig. 6?
L198-199 ‘The measured (points) and the predicted values agree excellently well’. This sentence needs to be removed.
Reviewer 2 Report
Suggestions for Authors:
The authors compare the feasibility of immobilized a-chymotrypsin enzyme regarding free enzyme in terms of reaction rate and kinetic. Moreover, they demonstrated the viability of immobilized enzyme in a biocatalytic membrane reactor. These are promising results, but some clarifications are necessary by the authors about the methodology in this work.
In the text, the authors used different enzyme concentration. For instance, page 3 of 9 and paragraph 112nd they wrote 1g/L of enzyme, then in figure 2 they wrote 1mg/L. Or in page 13 of 19 and paragraph 386st they wrote “applying 0.1 g/L enzyme concentration”. So, I consider that the authors should make it clear what was the reason for the chooses of different enzymatic concentration in different steps of the work, due to the enzyme concentration used can have a high influence on the results obtained. Please revised the manuscript and indicate in materials and methods section one paragraph where the readers can understand easier the enzyme concentration used in each step and the motivation for that.
As a suggestion, the authors did not demonstrate the reuse of immobilized enzyme, since this is the most important point regarding free enzyme, so I encourage the authors to continue in this line.
Reviewer 3 Report
Presented manuscript concerns immobilization of chemotrypsin using different strategies and characterization of the produced systems in various types of reactors. Presented approach might be interesting and some valuable information are presented. However, before manuscript might be considered for publication, it requires major revision. See some of the most important comments:
1 This title is good for review article. Tite should more refer to the content of the study.
2 Abstract is rather long and should be srhothned to be more consistent.
3 Why in the whole manuscript Authors use term "free enzyme immobilized". When enzyme is immobilized you can not use term free. Please carefully checked the whole manuscript to improve this issue.
4 The scheme for the used immobilization approahes should be added in the Introduction.
5 What was the substrate in reactions in FI\ig. 3.
6 Results section. What about data for NP1 in section 2.1? It should be added.
7 What was the repetition of the experiments? Error bars should be added.
8 The Authors shuld compare activity of free enzyme in both types of reactors and explain why free enzyme in memrane reactor retain higher activity for such longer period of time.
9 The novelty of the study has to be clearly presented in his manuscript as Authors have already published articles on similar topics.
10 Section 3.2. Beside equations presentation there is a lack of any discussion on the obtained data. Please improve this section and provide deeper discussion.
11 Line 293. Data mentioned in the text refer to another (not presented)data than values presented in Table 2.
12 Table 2 caption. I guess that instead of Reduction of, it should be retention of...
13 Line 320. This data re not presented in Table 2.
14 In the materials and methods section detail data has to be provided, to allow repetition of the experiments. For instance improve Section 4.2., everywhere add wavenumber value, add used concentration, etc...
15 How the kinetic paameters were calculated?? There is a lack of crucial data and experiments description.
16 The abstract and conclusions sections are almost the same. Conclusions should be rewritten to highlight the most important aspects of the study, its novelty and indicate possible future reserach in this area.
17 The whole manuscript has to be checked to avoid editorialand language issues.
18 Check and improve the references list to meet all of the Journal requirements.
Author Response
Reviewer 3
Comments and Suggestions for Authors
Presented manuscript concerns immobilization of chemotrypsin using different strategies and characterization of the produced systems in various types of reactors. Presented approach might be interesting and some valuable information are presented. However, before manuscript might be considered for publication, it requires major revision. See some of the most important comments:
1 This title is good for review article. Title should more refer to the content of the study.
Answer:
Original title: “Study of enzymatic biocatalytic reaction under different operation modes”;
Modified title: „Study of behavior of immobilized enzyme as enzyme nanoparticles and as biocatalytic membrane reactor”
2 Abstract is rather long and should be shortened to be more consistent.
Answer: Abstract has been shortened according to the following manner:
“Biocatalytic kinetic effect of a-chymotrypsin enzyme has been investigated in its free and pretreated (covered by a very thin, porous polymer layer, called enzyme nanoparticle) forms as well as its immobilized state of native form into pores of polysulfone/polyamide asymmetric, hydrophilic membrane. Trimethoxysilyl and acrylamide-bisacrylamide polymers were used for synthesis of enzyme nanoparticles. Applying Michaelis-Menten kinetics, the KM and vmax values of enzyme-polyacrylamide nanoparticles were about the same, as that of free enzyme. On the other hand, enzyme nanoparticles retain their activity 20-80 times longer time period than that of the free enzyme, but their initial activity values reduced to 13-55% of those of free enzymes, at 37 oC. Enzyme immobilized into asymmetric porous membrane layer remained active about 2.3 times longer time period than that of native enzyme (at pH = 7.4 and at 23 oC), while its reaction rate was about 8 times higher than that of free enzyme, measured it in mixed tank reactor. The conversion degree of substrate was gradually reduced in presence of increasing convective flux fluid phase. membrane reactor has transformed 2.5 times more substrate than the same amount of enzyme nanoparticles and 19 times more substrate than free enzyme, measured it in mixed tank reactor.”
3 Why in the whole manuscript Authors use term "free enzyme immobilized". When enzyme is immobilized you can not use term free. Please carefully checked the whole manuscript to improve this issue.
Answer: Whole manuscript was checked and this expression was corrected
“native enzyme, immobilized into porous membrane layer”
4 The scheme for the used immobilization approahes should be added in the Introduction.
Answer: Fig. 9. has been moved into the section 1 called Fig. 1.
5 What was the substrate in reactions in FI\ig. 3.
Answer: Figure capture was compared at the end: “ATEE was used as substrate both E and NP1.”
6 Results section. What about data for NP1 in section 2.1? It should be added.
Answer: Text was compared (L101-103): “For kinetic measurement NP2 was chosen because activity of NP1 was very low (about one order of magnitude less than activity of native enzyme. see also Table 2.) and it was hard to measure its activity values of same enzyme concentration values than native ones. Therefore, NP2 was chosen to study its kinetic values.”
7 What was the repetition of the experiments? Error bars should be added.
Answer: Error bars were added to the figures on Fig. 3., Fig. 5., Fig. 6., Fig. 7., and Fig. 8.
8 The Authors should compare activity of free enzyme in both types of reactors and explain why free enzyme in membrane reactor retains higher activity for such longer period of time.
Answer: “Half-life time of free enzyme is about 0.5 hour (see E on Figure 8), while that of NP2 is about 10 hours long, while the NP1 has the longest half-life time, about 42 hours as there are plotted in Fig. 7.”
9 The novelty of the study has to be clearly presented in his manuscript as Authors have already published articles on similar topics.
Answer: Conclusions section was compared with some novelties at the end(L484-487): “As the novelty of the presented results it was proved that preparation of the enzyme nanoparticles can significantly extend the active reaction time period of the enzyme providing much longer applicability. On the other hand the immobilization of native enzyme into porous membrane layer preparing a biocatalytic membrane reactor essentially increases of the bioreaction rate and it can work for longer time than that of the native enzyme.”
10 Section 3.2. Beside equations presentation there is a lack of any discussion on the obtained data. Please improve this section and provide deeper discussion.
Answer: Discussion was compared with the following considerations (L314-319): Namely, the value of the physical mass transfer coefficient of membrane, D/d=(1 x 10-9 m2/s)/200 x 10-6 m=0.5 x 10-5 m/s. The convective velocity was changed 1-15 x 10-5 m/s (Fig. 6.), these values are essentially higher practically in the whole convective flow regime. Accordingly the first term in Equation (1) can be neglected. According to this consideration, the mass transport equation (6) can be reduced to first-order differential equation:
11 Line 293. Data mentioned in the text refer to another (not presented)data than values presented in Table 2.
Answer: “(Table 2)” notification was removed from the end of the first paragraph of subchapter 3.3 (L332-335): “The above-mentioned three different immobilization methods of the α-chymotrypsin enzyme (NP1, NP2, MI) can be compared to each other, when one tries to calculate how much product is generated maximally by given amount of enzyme (e.g. 1 mg enzyme as free, E, and immobilized enzymes as, NP1, NP2, or MI).”
12 Table 2 caption. I guess that instead of Reduction of, it should be retention of...
Answer: The caption on Table 2. has been changed: “Reduction of biocatalytic activity of α-chymotrypsin enzyme during synthesis steps of two different synthesis methods (initial enzyme activities were considered as 100% in both cases). “
13 Line 320. This data are not presented in Table 2.
Answer: The number of Table has been changed from 2 to 3. “Similarly one can get effE, which equals to 1, effNP2 and effMI (Table 3).”
14 In the materials and methods section detail data has to be provided, to allow repetition of the experiments. For instance improve Section 4.2., everywhere add wavenumber value, add used concentration, etc...
Answer: Concentration values and other details are checked and corrected. An example was also added to the activity measurement (L437-439): “E.g. a typical reaction mixture was the following: 0.3 ml of 1 mg/L enzyme solution was mixed with 1.2 ml Tris/HCl buffer (80 mM Tris buffer, pH = 7.8) and with 1.5 ml 2 mM ATEE/Tris.”
15 How the kinetic parameters were calculated?? There is a lack of crucial data and experiments description.
Answer: The discussion was completed (L152-155): “According this method, reciprocal values of initial reaction rates (v0) are plotted as a function of reciprocal values of substrate concentrations. These values correlated linearly and the intercept of the fixed lines with vertical axis can get the reciprocal value of maximal velocity (1/vmax) and the cross-section with the horizontal axis can get the negative reciprocal value of the KM constant (-1/KM).”
16 The abstract and conclusions sections are almost the same. Conclusions should be rewritten to highlight the most important aspects of the study, its novelty and indicate possible future reserach in this area.
Answer: Conclusion section has been compared with highlights and future trends (L527-530): “Considering the future trends, it comes to the front to extend the active lifetime of enzyme in order to make its application more economic. Two main possibilities can be emphasized: the genetic modification of the enzyme as well as the elaboration of different preparation processes producing e.g. enzyme nanoparticles and biocatalytic membrane reactor as well as their combinations.”
17 The whole manuscript has to be checked to avoid editorial and language issues.
Answer: The language and expressions of the whole manuscript was checked and improved.
18 Check and improve the references list to meet all of the Journal requirements.
Answer: Reference list was also checked and modified.

Round 2
Reviewer 1 Report
The authors perfomed the suggested modifications, resulting in a moderately improved manuscript. However, some issues still remain: the title is still too generic. The enzyme used for the study should be mentioned in the title, since not all enzymes perform in the same manner during immobilization. Moreover, the parameters of SEM analysis are still not mentioned in the text (equipment used, beam current, voltage used). Error bars are missing from fig. 2.
The language was improved, but it is still my opinion that the manuscript should be revised by a professional language editor.
Author Response
COMMENT OF REVIEWER 1:
„The authors perfomed the suggested modifications, resulting in a moderately improved manuscript. However, some issues still remain: the title is still too generic. The enzyme used for the study should be mentioned in the title, since not all enzymes perform in the same manner during immobilization.”
ANSWER:
The title of the manuscript has been modified:
Original title: „Study of prepared enzymes as enzyme nanoparticles and of biocatalytic membrane reactor”
Modified title: „Study of prepared a-chymotrypsin as enzyme nanoparticles and of biocatalytic membrane reactor”
COMMENT OF REVIEWER 1:
„Moreover, the parameters of SEM analysis are still not mentioned in the text (equipment used, beam current, voltage used).”
ANSWER:
Description of SEM analysis has been modified:
Original text (L456-458 on P14 of original file): „Microstructure and pore size distribution of membranes were tested by scanning electron microscopy (FEI/ThermoFisher Apreo S). Surface of samples intended for SEM investigation was sputter coated by a 15 nm thick Au/Pd layer.”
Modified text: (Lines 577-581 in text with Track changes) “Microstructure and pore size distribution of membrane were tested by FEI/ThermoFisher Apreo S scanning electron microscope (SEM). This SEM investigation was realized in a low vacuum with an accelerator voltage of 5 kV. Membrane material is not conductive, therefore a thin electron transparent layer (about 15 nm thick Au/Pd layer) was prepared on the membrane surface by JEOL IB-29510VET type evaporator.”
COMMENT OF REVIEWER 1:
„Error bars are missing from figure 2.”
ANSWER:
Fig. 2 (which is Fig. 3. in the modified version) has been changed and error-bars are shown (see below).
Reviewer 2 Report
The authors have revised adequately the manuscript according to reviewer comments and given appropriate explanation for reviewer questions.
Author Response
Reviewer 2
Reviewer’s comment: The authors have revised adequately the manuscript according to reviewer comments and given appropriate explanation for reviewer questions.
Reply: authors warmly thank the acceptation of the revised version of the manuscript;
Reviewer 3 Report
I have carefully read the revised version of the manuscript. Authors have improved the manuscript significantly nd most of my querries have been properly explained. However, these is still one very important thong that has to be improved. In the previous revision, I have explained to the Authors, than if enzyme is immobilized you can not talk about free enzyme or you can not talk about native enzyme. When enzyme is immobilized you have to call it immobilized enzyme. This has to be improved in the whole manuscript.
Author Response
COMMENT OF REVIEWER 3:
„I have carefully read the revised version of the manuscript. Authors have improved the manuscript significantly nd most of my querries have been properly explained. However, these is still one very important thong that has to be improved. In the previous revision, I have explained to the Authors, than if enzyme is immobilized you can not talk about free enzyme or you can not talk about native enzyme. When enzyme is immobilized you have to call it immobilized enzyme. This has to be improved in the whole manuscript.”
ANSWER:
Enzyme that used in membrane reactor entrapped in membrane has not been prepared as enzyme nanoparticle (NP1 or NP2) before its immobilization. According to distinguish it, manuscript has been changed:
Original version (L102-105. on Page 3):
“Main aim of our research is to compare the efficiency and biocatalytic mechanism of two enzyme immobilization techniques, when the enzyme is used as its native form (abbreviated: E) or as enzyme nanoparticles (NP1: prepared by three-step synthesis method, and NP2: prepared by two-step synthesis method) and as the free enzyme is trapped into the membrane support layer (called MI).”
Corrected version: L124-128 with tracking: “Main aim of this research is to compare the efficiency and biocatalytic mechanism of two enzyme immobilization techniques, when the enzyme is used as its free form (abbreviated: E) or as enzyme nanoparticles (NP1: prepared by three-step synthesis method, and NP2: prepared by two-step synthesis method) and as the free (not immobilized, not pretreated as NP1 or NP2) enzyme (E) is trapped into the membrane support layer (called MI).”
Addition remark from authors: For better understanding or avoiding the misunderstanding the native attribute has been exchanged by free attribute of enzyme.
Round 3
Reviewer 1 Report
In this second review round, the authors performed the corrections requested by the reviewers, and thus the quality of the manuscript was significantly improved. In my opinion, the manuscript is now acceptable for publication, as long as one minor issue is addressed: In figures 3 and 6-8 the authors added the error bars (as requested by the reviewer) but they do not specify their meaning. For example, do they represent tha standard deviation or the standard error? From how many (technical or bioogical) replicates were they caculated? What do the x-axis error bars stand for? Please specify in the Materials and Methods section, or in the figure legends.
Author Response
ANSWERS TO REVIEWER 1.
Dear Reviewer 1,
authors thank your kind notifications
QUESTION AND COMMENT OF REVIWER 1.:
„In this second review round, the authors performed the corrections requested by the reviewers, and thus the quality of the manuscript was significantly improved. In my opinion, the manuscript is now acceptable for publication, as long as one minor issue is addressed: In figures 3 and 6-8 the authors added the error bars (as requested by the reviewer) but they do not specify their meaning. For example, do they represent tha standard deviation or the standard error? From how many (technical or bioogical) replicates were they caculated? What do the x-axis error bars stand for? Please specify in the Materials and Methods section, or in the figure legends.”
ANSWERS:
Figures and figure captions under figures 3., 4., 5., 6. 7. and 8. was corrected. During activity measurements and investigations of kinetics time was detected automatically by spectrophotometer, therefore standard error bars are as small as they are not shown on Figures 3. and 4. Error bars of time values on 7. and 8. are also very small.
Figure 3. on page 4. was replaced:
Time was detected automatically by spectrophotometer, no error bars have calculated. Figure caption under Figure 3. on page 4. was changed (L138-144):
ORIGINAL TEXT: „A) Substrate conversion (%) plotted as a function of time; B) Calculation of initial reaction rate values (vo) during the kinetic measurement. vo values can get as a slopes of lines which are plotted as linear regression of product concentration values as a function of time during the initial time period of biochemical reaction (e.g. during the first 2 minutes). Product concentration values are calculated from absorbance values of 3 ml reaction mixture in a quartz cuvette (with 1 mg/L α-chymotrypsin and at different substrate concentrations).”
CORRECTED TEXT: „A) Substrate conversion (%) plotted as a function of time; B) Calculation of initial reaction rate values (vo) during the kinetic measurement. vo values can get as a slopes of lines which are plotted as linear regression of product concentration values as a function of time during the initial time period of biochemical reaction (e.g. during the first 2 minutes). Product concentration values are calculated from absorbance values of 3 ml reaction mixture in a quartz cuvette (with 1 mg/L α-chymotrypsin and at different substrate concentrations). (Standard error values are plotted. They have varied between 4% up to 6%.)”
Figure 4. on page 5. was replaced:
Figure caption on Figure 4. on page 5. was also changed (L148-153 on modified text):
ORIGINAL TEXT: “Reciprocal values of initial reaction rate (v0) as a function of reciprocal substrate concentration obtained by free (not immobilized) α-chymotrypsin enzymes (E); and enzyme nanoparticles (NP2, prepared by two-step synthesis method) plotted by Lineweaver-Burk method. ATEE was used as substrate both for E and NP2.”
CORRECTED TEXT: “Reciprocal values of initial reaction rate (v0) as a function of reciprocal substrate concentration obtained by free (not immobilized) α-chymotrypsin enzymes (E); and enzyme nanoparticles (NP2, prepared by two-step synthesis method) plotted by Lineweaver-Burk method. ATEE was used as substrate both for E and NP2. (Standard error values of linear fitted data are plotted. It varies between ±20-25%. This relatively high errors are related to graphical method applied.)”
Figure 5 on page 6. was modified (error bars of x values were also plotted) and its figure capture was compared:
Figure 5.:
Figure caption of Figure 5. was also compared (L176-179 on text with track changes):
ORIGINAL TEXT: “Reaction rate values of biocatalytic membrane reactor in the function of convective flow across the membrane. Results show that the reaction rate is increasing by increase of the convective flow rate across the biocatalytic membrane layer.”
Compared text: “Reaction rate values of biocatalytic membrane reactor in the function of convective flow across the membrane. Results show that the reaction rate is increasing by increase of the convective flow rate across the biocatalytic membrane layer. (Standard error values are plotted. They have varied between ±4-6% of measured mean values.)”
Figure 6 on page 7. was modified and its figure capture was compared (x error bars (for Jw values) were also plotted):
Figure caption under Figure 6. was also compared (L176-179 on text with modifications):
ORIGINAL TEXT: „Change of the product concentration of enzyme catalytic membrane bioreactor (%) in the function of convective flow rate (Jw), applying different inlet substrate concentrations (0.5 mM ATEE, 1 mM ATEE and 2 mM ATEE). All curves could well describe by exponential functions with negative exponent (see continuous, tagged and dotted lines as calculated values) (Product concentration in % [cP] can be calculated as: cP = 100-cS).”
MODIFIED TEXT: „Change of the product concentration of enzyme catalytic membrane bioreactor (%) in the function of convective flow rate (Jw), applying different inlet substrate concentrations (0.5 mM ATEE, 1 mM ATEE and 2 mM ATEE). All curves could well describe by exponential functions with negative exponent (see continuous, tagged and dotted lines as calculated values) (Product concentration in % [cP] can be calculated as: cP = 100-cS). (Standard errors values are plotted. They have varied between ±4-6% of measured mean values.)”
Figure 7 on page 7. was modified:
Figure caption under Figure 7. was also compared (L207-211 on text with modifications):
ORIGINAL TEXT: “Activity changes of α-chymotrypsin enzyme nanoparticles covered by acrylamide-bisacrylamide random copolymer (NP2), or covered by poly(methacryloxypropyltrimethoxysilane) hybrid organic/inorganic polymer (NP1) and of free (not pretreated) enzyme (E) as a function of time measured in stirred tank reactor, with 150 rpm, at 37 oC.”
MODIFIED TEXT: “Activity changes of α-chymotrypsin enzyme nanoparticles covered by acrylamide-bisacrylamide random copolymer (NP2), or covered by poly(methacryloxypropyltrimethoxysilane) hybrid organic/inorganic polymer (NP1) and of free (not pretreated) enzyme (E) as a function of time measured in stirred tank reactor, with 150 rpm, at 37 oC. (Standard error values are plotted. They have varied between ±3.7% up to ±5.5%.)”
Figure 8 on page 9. was modified:
Figure caption under Figure 8. was also compared (L221-226 on text with modificatuions):
ORIGINAL TEXT:.”Reduction of activity of immobilized α-chymotrypsin enzyme into porous support layer of membrane (MI) compared to activity change of the not immobilized enzyme (E) measured at the same temperature (23 oC). Half-life times of E and MI was predicted from a scale and decay constants (λ) of E and MI, using equations (2) to (5). These calculated relative activity values are continuous line (E) and dash-line (MI). The measured (points) and the predicted values agree excellently well.”
MODIFIED TEXT: ”Reduction of activity of immobilized α-chymotrypsin enzyme into porous support layer of membrane (MI) compared to activity change of the not immobilized enzyme (E) measured at the same temperature (23 oC). Half-life times of E and MI was predicted from a scale and decay constants (λ) of E and MI, using equations (2) to (5). These calculated relative activity values are continuous line (E) and dash-line (MI). The measured (points) and the predicted values agree excellently well. Standard error values are plotted. They have varied between ±3.8% up to ±4.4%.)”
Text are corrected on p18 L438:
„Every measurement was repeated three times.”
and L446:
„Every measurement was repeated three times.”
